# Individual Privacy Accounting for Differentially Private Stochastic Gradient Descent

**Da Yu**                                                        *yuda3@mail2.sysu.edu.cn*
*Sun Yat-sen University*

**Gautam Kamath**[†] [*]                                          *g@csail.mit.edu*
*Cheriton School of Computer Science*
*University of Waterloo*

**Janardhan Kulkarni** [*]                                        *jakul@microsoft.com*
*Microsoft Research*

**Tie-Yan Liu** [*]                                               *tyliu@microsoft.com*
*Microsoft Research*

**Jian Yin** [*]                                                  *issjyin@mail.sysu.edu.cn*
*Sun Yat-sen University*

**Huishuai Zhang** [*]                                            *huzhang@microsoft.com*
*Microsoft Research*

**Reviewed on OpenReview:** *https://openreview.net/forum?id=l4JcxsOfpC*

## Abstract

Differentially private stochastic gradient descent (DP-SGD) is the workhorse algorithm for recent advances in private deep learning. It provides a single privacy guarantee to all datapoints in the dataset. We propose *output-specific* $(\varepsilon, \delta)$-DP to characterize privacy guarantees for individual examples when releasing models trained by DP-SGD. We also design an efficient algorithm to investigate individual privacy across a number of datasets. We find that most examples enjoy stronger privacy guarantees than the worst-case bound. We further discover that the training loss and the privacy parameter of an example are well-correlated. This implies groups that are underserved in terms of model utility simultaneously experience weaker privacy guarantees. For example, on CIFAR-10, the average $\varepsilon$ of the class with the lowest test accuracy is 44.2% higher than that of the class with the highest accuracy. Our code is available at `https://github.com/dayu11/individual_privacy_of_DPSGD`.

## 1 Introduction

Differential privacy is a strong notion of data privacy, enabling rich forms of privacy-preserving data analysis (Dwork et al., 2006; Dwork & Roth, 2014). Informally speaking, it quantitatively bounds the maximum influence of any datapoint using a privacy parameter $\varepsilon$, where smaller values of $\varepsilon$ correspond to stronger privacy guarantees. Training deep models with differential privacy is an active research area (Papernot et al., 2017; Zhu et al., 2020; Anil et al., 2021; Yu et al., 2022; Li et al., 2022; Golatkar et al., 2022; Mehta et al., 2022b; De et al., 2022; Bu et al., 2022; Mehta et al., 2022a). Models trained with differential privacy not only provide theoretical privacy guarantees to their data owners but also are more robust against empirical attacks (Rahman et al., 2018; Bernau et al., 2019; Carlini et al., 2019a; Jagielski et al., 2020; Nasr et al., 2021a).

Differentially private stochastic gradient descent (DP-SGD) is the most popular algorithm for differentially private deep learning (Song et al., 2013; Bassily et al., 2014; Abadi et al., 2016). At each step, DP-SGD takes

---

[†]Supported by an NSERC Discovery Grant, an unrestricted gift from Google, and a University of Waterloo startup grant.
[*]Authors are listed in alphabetical order.

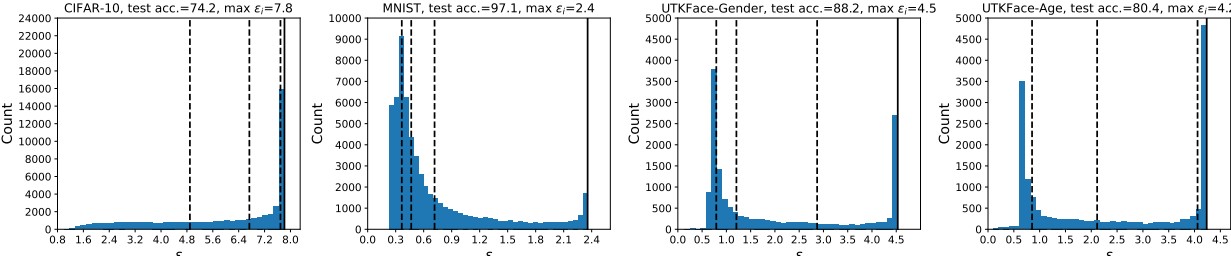

Figure 1: Individual privacy parameters of models trained by DP-SGD. The value of $\delta$ is $1 \times 10^{-5}$. The dashed lines indicate 30%, 50%, and 70% of datapoints. The black solid line shows the worst-case privacy parameter.

the model from the previous step and the dataset as inputs. It adds isotropic Gaussian noise to the average gradient of the current step. Models trained with DP-SGD satisfy $(\varepsilon, \delta)$-differential privacy. The canonical notion of differential privacy, including $(\varepsilon, \delta)$-DP, considers the *worst-case* privacy over all possible inputs. In the case of DP-SGD, this results in the privacy cost of all examples being computed with the largest possible magnitude of individual gradients, i.e., the gradient clipping threshold.

In practice, we may care more about the privacy guarantees of the models that will be deployed, which depend on the observed training trajectories. Broadly speaking, different examples may have very different impacts on a trained model (Feldman & Zhang, 2020; Jiang et al., 2021). Some examples may be easier to learn and hence the magnitudes of their individual gradients along the observed training trajectory could be much smaller than the clipping threshold. Such a fine-grained privacy guarantee can not be inferred by the canonical $(\varepsilon, \delta)$-DP because it requires the privacy guarantee to hold for all possible datasets and training trajectories. In this paper, we define *output-specific* $(\varepsilon, \delta)$-DP, which adapts to the training trajectory of the model to analyze the individual privacy of DP-SGD. Our definition captures the impact of various factors, such as the training set and algorithmic randomness, on individual privacy. We also develop algorithms to efficiently and accurately estimate individual privacy.

It turns out that, unsurprisingly, for common benchmarks, many examples experience much stronger privacy guarantees than implied by the worst-case DP analysis. To illustrate this, we plot the individual privacy parameters of four benchmark datasets in Figure 1. Experimental details, as well as more results, are in Sections 4 and 5. To the best of our knowledge, this paper is the first to reveal the disparity in individual privacy when running DP-SGD.

Further, we demonstrate a strong correlation between the privacy parameter of an example and its final training loss. That is, the examples with higher training loss also have higher privacy parameters in general. This suggests that the examples that suffer unfairness in terms of worse privacy are also the ones that have worse utility. See Figure 2 for an illustration. While prior works have shown that underrepresented groups experience worse utility (Buolamwini & Gebru, 2018), and that these disparities are amplified when models are trained privately (Bagdasaryan et al., 2019; Hansen et al., 2022; Noe et al., 2022; Lowy et al., 2022), we are the first to show that the privacy guarantee *and* utility are negatively impacted concurrently. In contrast, prior work that takes a worst-case perspective for privacy accounting, results in a uniform privacy guarantee for all training examples. For instance, when running gender classification on UTKFace, the average $\varepsilon$ of the race with the lowest test accuracy is 35.1% higher than that of the race with the highest accuracy.

## 1.1 Related Work

Several works have explored individual privacy analysis in differentially private learning. Jorgensen et al. (2015); Mühl & Boenisch (2022), and the work subsequent to ours of Boenisch et al. (2023), design learning algorithms that satisfy prespecified individual privacy parameters. Those prespecified parameters are independent of the learning algorithm, e.g., in some applications different users may have different expectations of privacy. Wang (2019) defines *Per-instance differential privacy* to analyze individual privacy when the target example is put in a fixed dataset. Redberg & Wang (2021) investigate per-instance DP of the objective perturbation algorithm

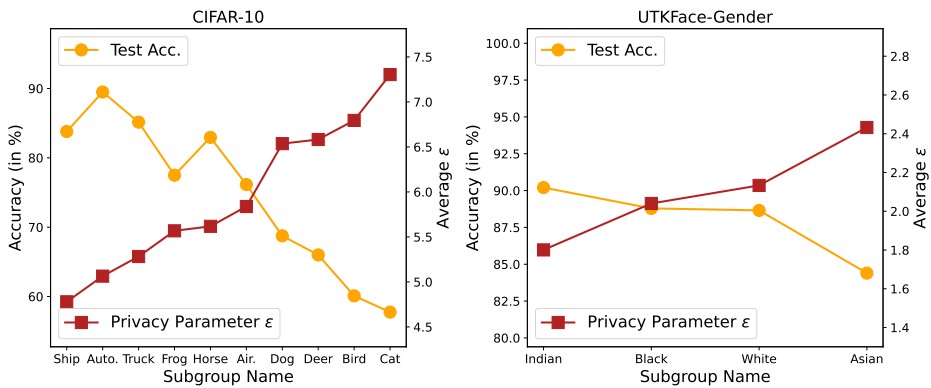

Figure 2: Accuracy and average $\varepsilon$ of different groups on CIFAR-10 and UTK-Face. Groups with worse accuracy also have worse privacy in general.

(Kifer et al., 2012). Golatkar et al. (2022) explore per-instance DP of differentially private batch gradient descent. Redberg & Wang (2021); Golatkar et al. (2022) focus on (strongly) convex objective functions because otherwise fixing the dataset is not sufficient to determine the individual privacy parameters. In this work, we define output-specific $(\varepsilon, \delta)$-DP that allows us to study the individual privacy of non-convex models trained by DP-SGD. We also conduct experiments on several datasets and demonstrate a strong correlation between privacy and utility.

Feldman & Zrnic (2021); Koskela et al. (2022) design individual *privacy filters* to make use of the variation in individual sensitivity. The filters allow examples with smaller per-step privacy costs to run for more steps until the accumulated cost reaches a target budget. By design, the target budget has to be a constant that is independent of the learning algorithm. Consequently, the formal privacy guarantee offered by the filters is still uniform for all examples.

Our output-specific $(\varepsilon, \delta)$-DP and the notion of *ex-post* DP by Ligett et al. (2017) both tailor the privacy guarantee to algorithm outcomes. Ex-post DP bounds the ratio between two probability/density functions at a single outcome. It can be generalized to pure differential privacy ($(\varepsilon, 0)$-DP). In contrast, DP-SGD uses Gaussian mechanisms and provides approximate differential privacy ($(\varepsilon, \delta)$-DP). There is no clean conversion between $(\varepsilon, \delta)$-DP and ex-post DP (Meiser, 2018). Therefore, our notion is necessary for analyzing individual privacy within the $(\varepsilon, \delta)$-DP framework.

## 2 Preliminaries

We first give some background on DP-SGD and explain why the canonical $(\varepsilon, \delta)$-DP is not suitable for measuring individual privacy. Then we define output-specific $(\varepsilon, \delta)$-DP. Finally, we give empirical evidence showing that providing the same privacy to all examples is not ideal.

### 2.1 Background on Differentially Private SGD

The privacy guarantee of DP-SGD is measured by $(\varepsilon, \delta)$-differential privacy.

**Definition 1.** *[$(\varepsilon, \delta)$-DP] An algorithm $\mathcal{A} : \mathcal{D} \to \mathcal{O}$ satisfies $(\varepsilon, \delta)$-differential privacy if for any pair of neighboring datasets $\mathbb{D}, \mathbb{D}' \in \mathcal{D}$ and any subset of outputs $\mathbb{S} \subset \mathcal{O}$ it holds that*

$$\Pr[\mathcal{A}(\mathbb{D}) \in \mathbb{S}] \le e^\varepsilon \Pr[\mathcal{A}(\mathbb{D}') \in \mathbb{S}] + \delta.$$

Two datasets $\mathbb{D}, \mathbb{D}'$ are neighboring datasets if they only differ in one datapoint. DP-SGD uses Rényi differential privacy (RDP) (Mironov, 2017) in privacy accounting to get a tighter composition bound (Abadi et al., 2016). After training, the accumulated RDP is converted to $(\varepsilon, \delta)$-DP. RDP measures the Rényi divergence at different orders. The Rényi divergence between two probability distributions $\mu$ and $\nu$ at order $\alpha$ is

$$D_\alpha(\mu||\nu) = \frac{1}{\alpha - 1} \log \int (\frac{d\mu}{d\nu})^\alpha d\nu.$$

Let $D_\alpha^\leftrightarrow(\mu||\nu) = \max(D_\alpha(\mu||\nu), D_\alpha(\nu||\mu))$ be the maximum divergence of two directions. The definition of RDP is as follows.

**Definition 2.** *[Rényi differential privacy (Mironov, 2017)] A randomized algorithm $\mathcal{A} : \mathcal{D} \to \mathcal{O}$ satisfies $(\alpha, \rho)$-RDP if for any neighboring datasets $\mathbb{D}, \mathbb{D}' \in \mathcal{D}$ it holds that*

$$D_\alpha^\leftrightarrow(\mathcal{A}(\mathbb{D})||\mathcal{A}(\mathbb{D}')) \le \rho.$$

When $\mathcal{A}$ is a deep learning algorithm, it is infeasible to directly measure the output distributions because of the non-convex nature of neural networks. To address this, DP-SGD makes each gradient update differentially private and uses the composition property of differential privacy to reason about the overall privacy cost.

**Definition 3.** *[Composition of RDP (Mironov, 2017)] Let $\mathcal{A}_1 : \mathcal{D} \to \mathcal{O}_1$ be $(\alpha, \rho_1)$-RDP and $\mathcal{A}_2 : \mathcal{O}_1 \times \mathcal{D} \to \mathcal{O}_2$ be $(\alpha, \rho_2)$-RDP, then the mechanism defined as $(X, Y)$, where $X \sim \mathcal{A}_1(\mathbb{D})$ and $Y \sim \mathcal{A}_2(X, \mathbb{D})$, satisfies $(\alpha, \rho_1 + \rho_2)$-RDP.*

The output of the composed algorithm is a tuple containing the outputs from all steps. Consequently, the output of DP-SGD at step $T$ is a sequence of models $(\theta_1, \theta_2, \ldots, \theta_T)$.

The privacy cost of a target example depends on its gradients along the training trajectory. We formalize the output distributions of DP-SGD at each step to illustrate this. DP-SGD uses Poisson sampling, i.e., each example is sampled independently with probability $p$. Let $\boldsymbol{v} = \sum_{i \in \mathbb{M}} \boldsymbol{g}_i$ be the sum of the minibatch gradients of $\mathbb{D}$, where $\mathbb{M}$ is the set of sampled indices. Consider also a neighboring dataset $\mathbb{D}'$ that has one datapoint $\boldsymbol{d}$ (with gradient $\boldsymbol{g}$) added. Because of Poisson sampling, the output is exactly $\boldsymbol{v}$ with probability $1 - p$ ($\boldsymbol{g}$ is not sampled) and is $\boldsymbol{v}' = \boldsymbol{v} + \boldsymbol{g}$ with probability $p$ ($\boldsymbol{g}$ is sampled). After adding isotropic Gaussian noise, the output distributions of two neighboring datasets are

$$\mathcal{A}(\mathbb{D}) \sim \mathcal{N}(\boldsymbol{v}, \sigma^2 \boldsymbol{I}). \tag{1}$$

$$\mathcal{A}(\mathbb{D}') \sim \mathcal{N}(\boldsymbol{v}, \sigma^2 \boldsymbol{I}) \text{ with prob. } 1 - p,$$

$$\mathcal{A}(\mathbb{D}') \sim \mathcal{N}(\boldsymbol{v}', \sigma^2 \boldsymbol{I}) \text{ with prob. } p. \tag{2}$$

The RDP of $\boldsymbol{d}$ at the current step is the Rényi divergences between Equation (1) and (2). For a given $\sigma$ and $p$, the divergences are determined by the $L_2$ norm of $\boldsymbol{g} = \boldsymbol{v}' - \boldsymbol{v}$. Therefore, a larger gradient would result in a larger privacy cost. If we consider all possible $\theta_{t-1} \in \mathcal{O}_{t-1}$ as required by Definition 1, we would have to compute the divergences with the largest possible magnitude of $\boldsymbol{g}$. Abadi et al. (2016) use the gradient clipping threshold to compute the privacy cost, which results in a worst-case privacy analysis for every example.

### 2.2 Output-specific $(\varepsilon, \delta)$-Differential Privacy

We define output-specific individual $(\varepsilon, \delta)$-differential privacy to provide a fine-grained analysis of individual privacy. It makes the privacy parameter $\varepsilon$ a function of the outputs and the target datapoint.

**Definition 4.** *[Output-specific individual $(\varepsilon, \delta)$-DP] Fix a datapoint $\boldsymbol{d}$ and a set of outcomes $\mathbb{A} \subset \mathcal{O}$. Let $\mathbb{D}$ be an arbitrary dataset and $\mathbb{D}' = \mathbb{D} \cup \{\boldsymbol{d}\}$, an algorithm $\mathcal{A} : \mathcal{D} \to \mathcal{O}$ satisfies output-specific individual $(\varepsilon(\mathbb{A}, \boldsymbol{d}), \delta)$-DP for $\boldsymbol{d}$ at $\mathbb{A}$ if for any $\mathbb{S} \subset \mathbb{A}$*

$$\Pr[\mathcal{A}(\mathbb{D}) \in \mathbb{S}] \le e^{\varepsilon(\mathbb{A}, \boldsymbol{d})} \Pr[\mathcal{A}(\mathbb{D}') \in \mathbb{S}] + \delta,$$

$$\Pr[\mathcal{A}(\mathbb{D}') \in \mathbb{S}] \le e^{\varepsilon(\mathbb{A}, \boldsymbol{d})} \Pr[\mathcal{A}(\mathbb{D}) \in \mathbb{S}] + \delta.$$

Definition 4 is a strict generalization of $(\varepsilon, \delta)$-DP as one can recover $(\varepsilon, \delta)$-DP by maximizing $\varepsilon(\mathbb{A}, \boldsymbol{d})$ over $\mathbb{A}$ and $\boldsymbol{d}$. With $\varepsilon$ being a function of some outcomes, we can analyze the individual privacy of models trained by DP-SGD by fixing the first $T - 1$ models $(\theta_1, \ldots, \theta_{T-1})$, which fully specify the gradients along the training. Such a fine-grained analysis is meaningful because the privacy risk of deploying a specific model that is always bound to an observed training trajectory.

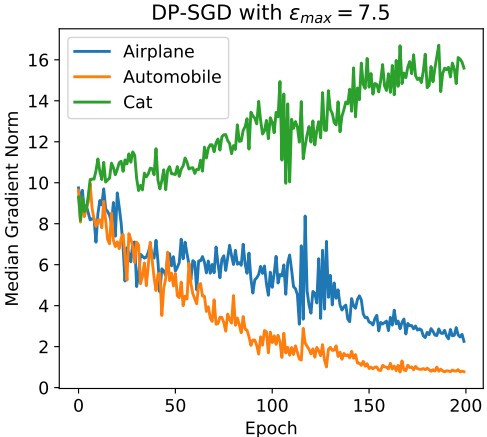

Figure 3: Median of gradient norms of different classes when training a ResNet-20 model on CIFAR-10.

### 2.3 Gradients of Different Examples Vary Significantly

At each step of DP-SGD, the privacy cost of an example depends on its gradient at the current step (see Section 2.1 for details). In this section, we empirically show gradients of different examples vary significantly to demonstrate that different examples experience very different privacy costs. We train a ResNet-20 model with DP-SGD on CIFAR-10. The maximum clipping threshold is the median of gradient norms at initialization. More implementation details are in Section 4. We plot the median of gradient norms of three different classes in Figure 3. The gradient norms of different classes show significant stratification. Such stratification naturally leads to different privacy costs. This suggests that it is meaningful to further quantify individual privacy parameters.

## 3 Individual Privacy of DP-SGD

Algorithm 1 shows the implementation of DP-SGD[1] (Abadi et al., 2016). Theorem 3.1 gives the individual privacy analysis of DP-SGD. Algorithm 2 gives the pseudocode for computing individual privacy parameters. At each step, Algorithm 2 uses the (estimated) individual gradient norms to compute per-step Rényi differential privacy (RDP) for every example. It also updates the individual gradient norms and the accumulated RDP. We introduce two arguments in Algorithm 2 to reduce the computational cost of individual privacy accounting. The first one is the frequency $K$ of computing batch gradient norms and the second one is whether to round individual gradient norms with a small constant $r$. More discussion on these two arguments could be found in Section 3.1 and 3.2.

**Theorem 3.1.** *Let $\{\theta_1, \ldots, \theta_{t-1}\}$ be the observed models at step $t$. Suppose we run Algorithm 1 with $K = 1$ and without rounding, then Algorithm 1 satisfies $(o_\alpha^{(i)} + \frac{\log(1/\delta)}{\alpha - 1}, \delta)$-output-specific individual DP for the $i_{th}$ example at $\mathbb{A} = (\theta_1, \ldots, \theta_{t-1}, \mathcal{O}_t)$, where $o_\alpha^{(i)}$ is the accumulated RDP at order $\alpha$ and $\mathcal{O}_t$ is the range of $\mathcal{A}_t$.*

*Proof Sketch.* At step $t$, given the observed models $(\theta_1, \ldots, \theta_{t-1})$, the composited training algorithm is

$$\hat{\mathcal{A}}^{(t)} = (\mathcal{A}_1(\mathbb{D}), \mathcal{A}_2(\theta_1, \mathbb{D}), \ldots, \mathcal{A}_t(\theta_1, \ldots, \theta_{t-1}, \mathbb{D})).$$

We first use the composition theorem to show the accumulated RDP is the RDP of $\hat{\mathcal{A}}^{(t)}$. Then we prove the RDP bound on $\hat{\mathcal{A}}^{(t)}$ gives an output-specific-$(\varepsilon, \delta)$-DP bound on Algorithm 1. We relegate the proof to Appendix A. □

---

[1]Our implementation of DP-SGD follows the privacy analysis in Abadi et al. (2016) which uses Poisson sampling. We note that many existing implementations of DP-SGD use shuffle data instead of Poisson sampling to enforce stochasticity. Shuffle data is easier to implement but using it would create a mild discrepancy with the analysis in Abadi et al. (2016). Formal privacy analysis of shuffle data requires *privacy amplification by shuffling* (Koskela et al., 2023; Wang, 2023; Feldman et al., 2023).

---

**Algorithm 1** Differentially Private SGD

---

**Input:** Clipping threshold $C$, noise variance $\sigma^2$, sampling probability $p$, number of steps $T$.

Let $\{Z^{(i)} = C\}_{i=1}^n$ be the estimates of individual gradient norms, initialized as $C$.

Let $\{\boldsymbol{o}^{(i)} = 0\}_{i=1}^n$ be the accumulated individual RDP.

**for** $t = 0$ *to* $T - 1$ **do**

    *//Individual privacy accounting.*

    Run Algorithm 2 with $\{Z^{(i)}\}_{i=1}^n$ and $\{\boldsymbol{o}^{(i)}\}_{i=1}^n$.

    Update $\{Z^{(i)}\}_{i=1}^n$ and $\{\boldsymbol{o}^{(i)}\}_{i=1}^n$ with the results of Algorithm 2.

    *//Run DP-SGD as usual.*

    Sample a minibatch of gradients $\{\boldsymbol{g}^{(I_j)}\}_{j=1}^{|I|}$ with probability $p$ , where $I$ is the sampled indices.

    Clip gradients $\bar{\boldsymbol{g}}^{(I_j)} = clip(\boldsymbol{g}^{(I_j)}, C)$.

    Update model $\theta_t = \theta_{t-1} - \eta(\sum \bar{\boldsymbol{g}}^{(I_j)} + z)$, where $z \sim \mathcal{N}(0, \sigma^2 \boldsymbol{I})$.

**end for**

---

**Algorithm 2** Individual Privacy Accounting for DP-SGD

---

**Input:** Individual gradient norms $\{Z^{(i)}\}_{i=1}^n$, accumulated individual RDP $\{\boldsymbol{o}^{(i)}\}_{i=1}^n$, frequency $K$ of updating $\{Z^{(i)}\}_{i=1}^n$, rounding precision $r$, current iteration $t$.

**if** $t \bmod K = 0$ **then**

    *//Update individual sensitivity.*

    Compute batch gradient norms $\{\left\|\boldsymbol{g}^{(i)}\right\|_2\}_{i=1}^n$.

    Update $Z^{(i)} = \min(\left\|\boldsymbol{g}^{(i)}\right\|_2, C)$.

    **if** use rounding **then**

        *//Reduce the number of different norms.*

        Update $\{Z^{(i)} = \arg\min_{c \in \mathbb{C}}(|c - Z^{(i)}|)\}_{i=1}^n$, where $\mathbb{C} = \{r, 2r, \ldots, C\}$ contains all possible norms.

    **end**

**end**

*//Compute the current step RDP.*

Compute the Rényi divergences between Equation 1 and 2 numerically with $Z_i$, $p$, and $\sigma^2$ and store the result in $\boldsymbol{\rho}^{(i)}$.

*//Update the accumulated RDP.*

$\boldsymbol{o}^{(i)} = \boldsymbol{o}^{(i)} + \boldsymbol{\rho}^{(i)}$.

**return** $\{Z^{(i)}\}_{i=1}^n$, $\{\boldsymbol{o}^{(i)}\}_{i=1}^n$

---

**Remark 1.** *Algorithm 2 does not change the worst-case privacy guarantee (Definition 1) of DP-SGD because it does not modify the update rule.*

In Theorem 3.1, we run Algorithm 2 with $K = 1$ and without rounding individual gradient norms. This configuration is computationally expensive for two reasons. Firstly, setting $K = 1$ requires computing batch gradient norms at each SGD update. Secondly, the number of unique gradient norms is large without rounding. Each unique gradient norm corresponds to a different single-step RDP that needs to be computed numerically. In Section 3.1, we give more details on the computational challenges. In Section 3.2, we use a larger $K$ and round individual gradient norms to provide estimations of individual privacy. This greatly improves the efficiency of Algorithm 2. In Section 3.3, we show the estimates of individual privacy parameters are accurate.

### 3.1 Computational Challenges of Individual Privacy

The first challenge is that computing exact privacy costs at each step requires batch gradient norms, which is impractical when running SGD. At each update, we need the gradient of every example to compute the corresponding RDP between Equation 1 and 2. The worst-case privacy analysis of DP-SGD does not have this problem because it simply assumes all examples have the maximum possible gradient, i.e., the clipping threshold.

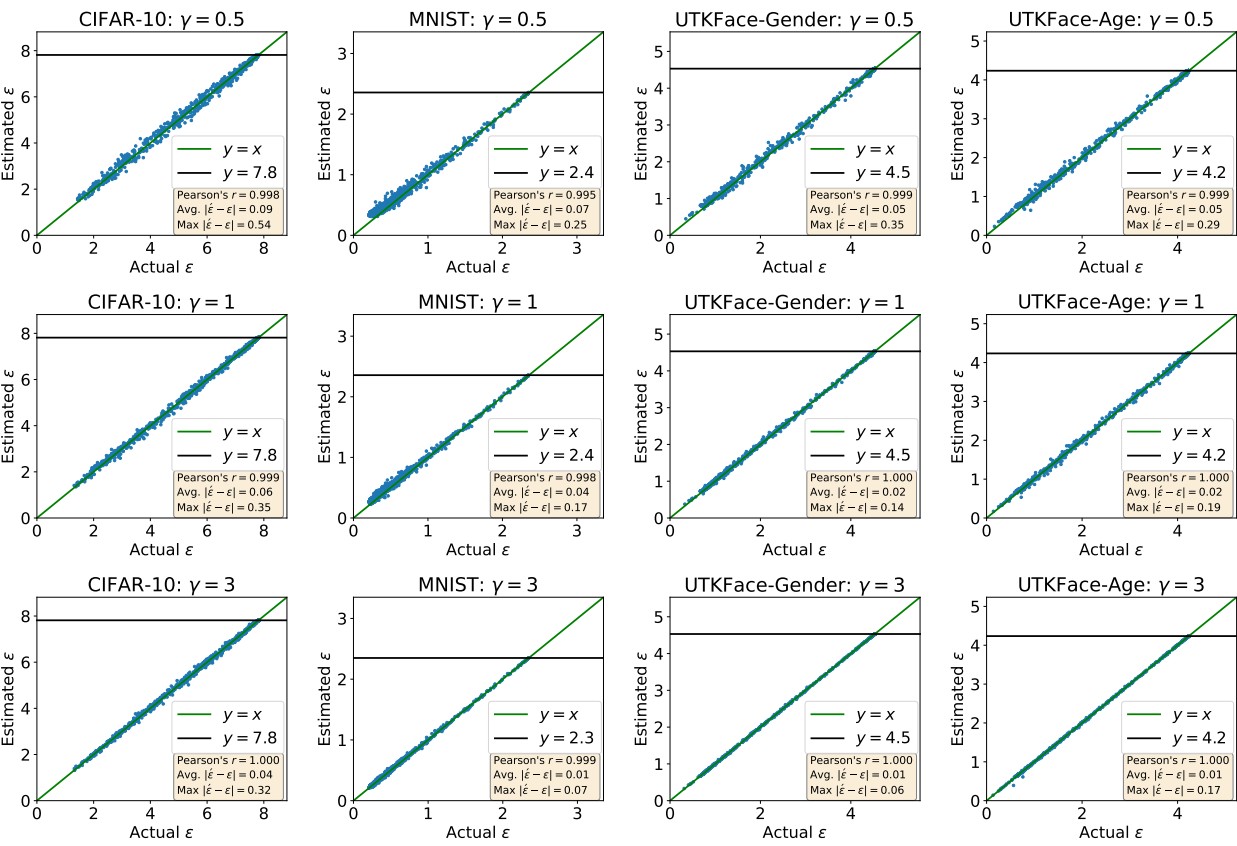

Figure 4: Privacy parameters based on estimations of individual gradient norms ($\varepsilon$) versus those based on exact ones ($\acute{\varepsilon}$). The value of $\gamma$ denotes the number of updates of full gradient norms per epoch. The horizontal line shows the worst-case privacy guarantee.

The second challenge is that the RDP of every example at each step needs to be computed numerically. It has been shown that numerical computations are necessary to get tight bounds on the Rényi divergences between Equation 1 and 2 (Abadi et al., 2016; Wang et al., 2019; Mironov et al., 2019; Gopi et al., 2021). In Abadi et al. (2016), only one numerical computation is required because all examples are assumed to have the worst-case privacy cost. However, when computing individual privacy parameters, the number of numerical computations is the same as the number of different gradients that could be as large as $n \times T$, where $n$ is the dataset size and $T$ is the number of iterations.

## 3.2 Improving the Efficiency of Individual Privacy Accounting

We only compute the batch gradient norms every $K$ iteration to reduce the computational overhead. The norms are then used to estimate the privacy costs for the subsequent iterations. We note that providing estimates of privacy costs is inevitable when the computational budget is limited. This is because, by the nature of SGD, one does not have the exact individual gradient norms at every iteration. In Appendix B.2, we explore another design choice which is to clip $\boldsymbol{g}^{(i)}$ with $Z^{(i)}$ in Algorithm 1. Although this slightly changes the implementation of DP-SGD, Algorithm 2 would return the exact privacy costs. We run experiments with this design choice and report the results in Appendix B.2. Our observations in the main text still hold in Appendix B.2.

To reduce the number of numerical computations, we round individual gradient norms with a small constant $r$. Because the maximum clipping threshold $C$ is a constant, then, by the pigeonhole principle, there are at most $\lceil C/r \rceil$ different values of gradient norms, and hence there are at most $\lceil C/r \rceil$ different values of RDP between Equation (1) and (2). Note that $r$ should be small enough to avoid underestimation of RDP. We set $r = 0.01C$ throughout this paper.

Table 1: Computational costs of computing individual privacy parameters for CIFAR-10.

|  | w/ rounding | w/o rounding |
|---|---|---|
| # of computations | $1 \times 10^2$ | $1 \times 10^7$ |
| Time (in seconds) | $< 3$ | $\sim 2.6 \times 10^4$ |

We compare the computational costs with/without rounding in Table 1. We run the numerical method in Mironov et al. (2019) once for every different value of RDP (with the default setup in the Opacus library (Yousefpour et al., 2021a)). We run DP-SGD on CIFAR-10 for 200 epochs. The full gradient norms are updated once per epoch. All results in Table 1 use multiprocessing with 5 cores of an AMD EPYC™ 7V13 CPU. With rounding, the overhead of computing individual privacy parameters is negligible. The computational cost without rounding is more than 7 hours.

### 3.3 Estimates of Individual Privacy Are Accurate

We run Algorithm 2 with the setup in Section 3.2 and compare the results with ground-truth values. To compute the ground-truth individual privacy, we randomly sample 1000 examples before training. During training, we compute the exact privacy costs for the same 1000 examples at every iteration.

We compute the Pearson correlation coefficient between the estimations and the ground-truth values. We also compute the average and the worst absolute errors. We report results on MNIST, CIFAR-10, and UTKFace. Details about the experiments are in Section 4. We plot the results in Figure 4. The estimations of $\varepsilon$ are close to those ground-truth values (Pearson's $r > 0.99$) even when we only update the gradient norms every two epochs ($\gamma = 0.5$). Updating batch gradient norms more frequently further improves the estimation, though doing so would increase the computational overhead.

It is worth noting that the maximum clipping threshold $C$ affects the computed privacy parameters. Large $C$ increases the variation of gradient norms (and hence the variation of privacy parameters) but leads to large noise variance while small $C$ suppresses the variation and leads to large gradient bias. Large noise variance and gradient bias are both harmful to learning (Chen et al., 2020; Song et al., 2021). In Appendix C, we show the influence of using different $C$ on both accuracy and privacy.

### 3.4 What Can We Do with Individual Privacy Parameters?

Individual privacy parameters depend on the private data and are thus sensitive. They can not be released publicly without care. We describe some approaches to safely make use of individual privacy parameters.

The first approach is to release $\varepsilon_i$ to the owner of $\boldsymbol{d}_i$. This approach does not incur additional privacy cost for two reasons. First, it is safe for $\boldsymbol{d}_i$ because only the rightful owner sees $\varepsilon_i$. Second, releasing $\varepsilon_i$ does not increase the privacy cost of any other example $\boldsymbol{d}_j \neq \boldsymbol{d}_i$. This is because computing $\varepsilon_i$ can be seen as a post-processing of $(\theta_1, \ldots, \theta_{t-1})$, which is reported in a privacy-preserving manner. We prove the claim in Theorem 3.2.

**Theorem 3.2.** *Let $\mathcal{A} : \mathcal{D} \to \mathcal{O}$ be an algorithm that is $(\varepsilon_j, \delta)$-output-specific individual DP for $\boldsymbol{d}_j$ at $\mathbb{A} \subset \mathcal{O}$. Let $f(\cdot, \boldsymbol{d}_i) : \mathcal{O} \to \mathcal{R} \times \mathcal{O}$ be a post-processing function that returns the privacy parameter of $\boldsymbol{d}_i$ ($\neq \boldsymbol{d}_j$) and the training trajectory. We have $f(\cdot, \boldsymbol{d}_i)$ is $(\varepsilon_j, \delta)$-output-specific DP for $\boldsymbol{d}_j$ at $\mathbb{F} \subset \mathcal{R} \times \mathcal{O}$ where $\mathbb{F} = \{f(a, \boldsymbol{d}_i) : a \in \mathbb{A}\}$ is all possible post-processing results.*

*Proof.* First note that the construction of $f(\cdot, \boldsymbol{d}_i)$ does not increase the privacy cost of $\boldsymbol{d}_j$ because it is independent of $\boldsymbol{d}_j$. Without loss of generality, let $\mathbb{D}, \mathbb{D}' \in \mathcal{D}$ be the neighboring datasets where $\mathbb{D}' = \mathbb{D} \cup \{\boldsymbol{d}_j\}$. Let $\mathbb{S} \subset \mathbb{F}$ be an arbitrary event and $\mathbb{T} = \{a \in \mathbb{A} : f(a, \boldsymbol{d}_i) \in \mathbb{S}\}$. Because $f$ is a bijective function, we have

$$\Pr\left[f\left(\mathcal{A}(D), \boldsymbol{d}_i\right) \in \mathbb{S}\right] = \Pr\left[\mathcal{A}(D) \in \mathbb{T}\right] \tag{3}$$

$$\leq e^{\varepsilon_j} \Pr\left[\mathcal{A}(D') \in \mathbb{T}\right] + \delta \tag{4}$$

$$= e^{\varepsilon_j} \Pr\left[f\left(\mathcal{A}(D'), \boldsymbol{d}_i\right) \in \mathbb{S}\right] + \delta, \tag{5}$$

Table 2: Statistics of individual privacy parameters can be accurately released with minor privacy costs. The average estimation error rate is 1.13% for MNIST and 0.91% for CIFAR-10. The value of $\delta$ is $1 \times 10^{-5}$.

| MNIST | Average | 0.1-quantile | 0.3-quantile | Median | 0.7-quantile | 0.9-quantile |
|---|---|---|---|---|---|---|
| Non-private | 0.686 | 0.236 | 0.318 | 0.431 | 0.697 | 1.682 |
| $\varepsilon = 0.1$ | 0.681 | 0.238 | 0.317 | 0.436 | 0.708 | 1.647 |
| CIFAR-10 | Average | 0.1-quantile | 0.3-quantile | Median | 0.7-quantile | 0.9-quantile |
| Non-private | 5.942 | 2.713 | 4.892 | 6.730 | 7.692 | 7.815 |
| $\varepsilon = 0.1$ | 5.939 | 2.801 | 4.876 | 6.744 | 7.672 | 7.923 |

which completes the proof. Using a bijective post-processing function is necessary for Theorem 3.2 to hold. Otherwise, there may be some $o \notin \mathbb{A}$ and $a \in \mathbb{A}$ that have the same processed output, which invalids the derivation from Equation 4 to Equation 5. $\qquad \square$

The second approach is to privately release aggregate statistics of the population, e.g., the average or quantiles of the $\varepsilon$ values. Recent works have demonstrated such statistics can be published accurately with a minor privacy cost Andrew et al. (2021). Specifically, we privately release the average and quantiles of the $\varepsilon$ values. We report the results on CIFAR-10 and MNIST. For releasing the average value of $\varepsilon$, we use the Gaussian Mechanism. For releasing the quantiles, we use 20 steps of batch gradient descent to solve the objective function in Andrew et al. (2021) with the default setup. The results are in Table 2. The released statistics are close to the actual values under $(0.1, 10^{-5})$-DP.

Finally, individual privacy parameters can also serve as a powerful tool for a trusted data curator to improve the model quality. By analyzing the individual privacy parameters of a dataset, a trusted curator can focus on collecting more data representative of the groups that have higher privacy risks to mitigate the disparity in privacy.

## 4 Individual Privacy Parameters on Different Datasets

In Section 4.1, we first show the distribution of individual privacy parameters on four tasks. Then we study how individual privacy parameters correlate with training loss in Section 4.2. The experimental setup is as follows.

**Datasets.** We use two benchmark datasets MNIST ($n = 60000$) and CIFAR-10 ($n = 50000$) (LeCun et al., 1998; Krizhevsky, 2009) as well as the UTKFace dataset ($n \simeq 15000$) (Zhang et al., 2017) that contains the face images of four different races (White, $n \simeq 7000$; Black, $n \simeq 3500$; Asian, $n \simeq 2000$; Indian, $n \simeq 2800$). We construct two classification tasks on UTKFace: predicting gender, and predicting whether the age is under 30.[2] We slightly modify the dataset between these two tasks by randomly removing a few examples to ensure each race has balanced positive and negative labels.

**Models and hyperparameters.** For CIFAR-10, we use the WRN16-4 model in De et al. (2022), which achieves advanced performance in private setting. We follow the implementation details in De et al. (2022) expect their data augmentation method to reduce computational cost. For MNIST and UTKFace, we use ResNet20 models with batch normalization layers replaced by group normalization layers. For UTKFace, we initialize the model with weights pre-trained on ImageNet.

We set $C = 1$ on CIFAR-10, following De et al. (2022). For MNIST and UTKFace, we set $C$ as the median of gradient norms at initialization, following the suggestion in Abadi et al. (2016). The privacy cost of using the median is not taken care of. However, the median of gradient norms could be released accurately with a small privacy cost[3]. The batchsize is 4096 for CIFAR-10 and 1024 for MNIST and UTKFace. The training epoch is 300 for CIFAR-10 and 100 for MNIST and UTKFace. For a target maximum $\varepsilon$, we use the package

---

[2]We acknowledge that predicting gender and age from images may be problematic. Nonetheless, as facial images have previously been highlighted as a setting where machine learning has disparate accuracy on different groups, we revisit this domain through a related lens. The labels are provided by the dataset curators.

[3]For instance, if we use the algorithm in Andrew et al. (2021) to privatize the median gradient norm of the UTKFace-Gender dataset with ($\varepsilon = 0.1, \delta = 1 \times 10^{-5}$). The non-private median is 15.73 and the privatized median is 15.82.

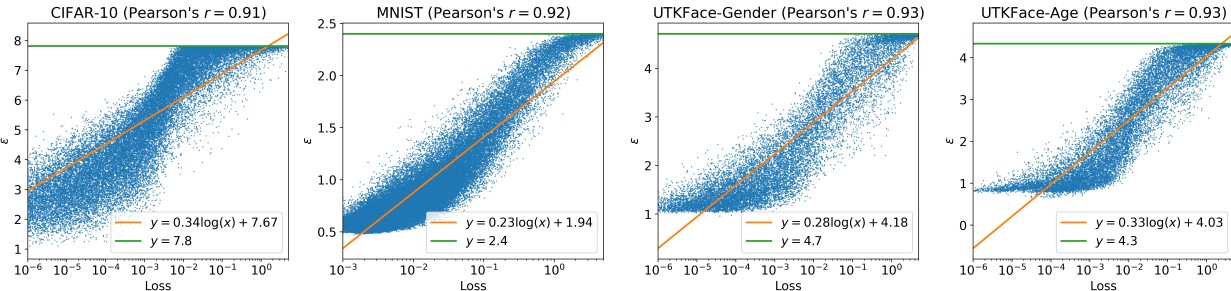

Figure 5: Privacy parameters and final training losses. Each point shows the final training loss and privacy parameter of one example. Pearson's $r$ is computed between privacy parameters and log loss values.

in the Opacus library to find the corresponding noise variance Yousefpour et al. (2021b). We update the batch gradient norms three times per epoch for all experiments in this section (the case of $\gamma = 3$ in Figure 4). All experiments are run on single Tesla V100 GPUs with 32G memory. Our source code will be publicly available.

### 4.1 Individual Privacy Parameters Vary Significantly

Figure 1 shows the individual privacy parameters on all datasets. The privacy parameters vary across a large range on all four tasks. On the CIFAR-10 dataset, the maximum $\varepsilon_i$ is 7.8 while the minimum $\varepsilon_i$ is 1.0. When running gender classification on the UTKFace dataset, the maximum $\varepsilon_i$ is 4.5 while the minimum $\varepsilon_i$ is only 0.1.

We also observe that, for easier tasks, more examples enjoy stronger privacy guarantees. For example, $\sim35\%$ of examples reach the worst-case $\varepsilon$ on CIFAR-10 while only $\sim3\%$ do so on MNIST. This may be because the loss decreases quickly when the task is easy, resulting in gradient norms also decreasing and thus stronger privacy guarantees.

### 4.2 Privacy Parameters and Loss Are Positively Correlated

We study how individual privacy parameters correlate with final training loss values. The privacy parameter of one example depends on its gradient norms along the training. In strongly convex optimization, the loss value of an example is reflected in the norm of its gradient. However, for non-convex deep models, there is no clear relation between the final training loss of one example and its gradient norms. Therefore, we run experiments to reveal the empirical correlation between privacy and utility.

We visualize individual privacy parameters and the final training loss values in Figure 5. The individual privacy parameters increase with loss until they reach the maximum $\varepsilon$. To quantify the order of correlation, we further fit the points with one-dimensional logarithmic functions and compute the Pearson correlation coefficients between the privacy parameters and log loss values. The Pearson correlation coefficients are larger than 0.9 on all datasets, showing a logarithmic correlation between the privacy parameter of a datapoint and its final training loss. In Appendix E, we experiment with $C = 5$ and $C = 10$ on CIFAR-10 to study the correlation between training loss and privacy under various clipping thresholds. The Pearson correlation coefficients are 0.89 and 0.9 for $C = 5$ and $C = 10$, respectively, suggesting that there is still a positive logarithmic correlation.

## 5 Groups Are Simultaneously Underserved in Both Accuracy and Privacy

It is well-documented that the accuracy of machine learning models may be unfair for different subpopulations (Buolamwini & Gebru, 2018; Bagdasaryan et al., 2019; Suriyakumar et al., 2021). Our finding demonstrates that this disparity may be simultaneous in terms of both accuracy *and* privacy. We empirically verify this by plotting the average $\varepsilon$ and test accuracy of different groups. The experiment setup is the same as Section 4. For CIFAR-10 and MNIST, the groups are the data from different classes, while for UTKFace, the groups are the data from different races.

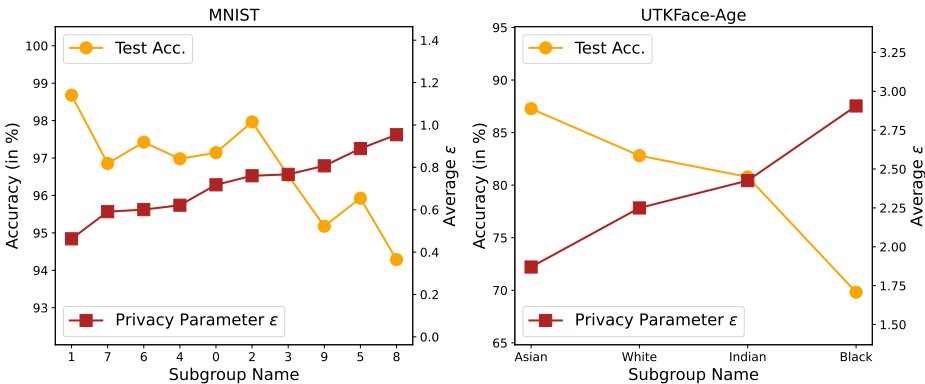

Figure 6: Accuracy and average $\varepsilon$ of different groups on MNIST and UTK-Age. Groups with worse accuracy also have worse privacy in general.

We plot the results in Figure 2 and 6. The groups are sorted based on the average $\varepsilon$. The test accuracy of different groups correlates well with the average $\varepsilon$ values. Groups with worse accuracy do have worse privacy guarantees in general. On CIFAR-10, the average $\varepsilon$ of the 'Cat' class (which has the worst test accuracy) is 44.2% higher than the average $\varepsilon$ of the 'Automobile' class (which has the highest test accuracy). On UTKFace-Gender, the average $\varepsilon$ of the group with the lowest test accuracy ('Asian') is 35.1% higher than the average $\varepsilon$ of the group with the highest accuracy ('Indian'). Similar observation also holds on other tasks. To the best of our knowledge, our work is the first to reveal this simultaneous disparity. In Appendix D, we run membership inference attacks to show the disparity in privacy parameters reflects the disparity in empirical privacy risks.

## 6 Conclusion

We define output-specific individual $(\varepsilon, \delta)$-differential privacy to characterize the individual privacy guarantees of models trained by DP-SGD. We also design an efficient algorithm to accurately estimate the individual privacy parameters. We use this new algorithm to examine individual privacy guarantees on several datasets. Significantly, we find that groups with worse utility also suffer from worse privacy. This new finding reveals the complex while interesting relation among utility, privacy, and fairness. It suggests that mitigating the utility fairness under differential privacy is more tricky than doing so in the non-private case. This is because classic methods such as upweighting underserved examples would exacerbate the disparity in privacy. We hope that our work sheds new light on this timely topic.

## Broader Impact

One way to utilize individual privacy parameters is to release them to corresponding users (see Section 3.4 for details). This provides a more accurate, and hence more responsible, privacy report. However, releasing individual privacy parameters may pose new challenges when a machine learning system enables data deletion, also known as machine unlearning (Ginart et al., 2019; Bourtoule et al., 2019). Data deletions may worsen the privacy guarantees of the remaining examples (Carlini et al., 2022). Moreover, groups with larger $\varepsilon$ may send deletion requests more frequently than others, which could further deteriorate their privacy guarantees (Hashimoto et al., 2018). It's important to keep these considerations in mind when implementing both data deletion and individual privacy accounting in real-world settings.

## Acknowledgments

The authors express their gratitude to Yu-Xiang Wang and Saeed Mahloujifar for their valuable comments on an earlier version of this paper, and to the anonymous reviewers for their insightful feedback.

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

# A    Proof of Theorem 3.1

**Theorem 3.1.** *Let $\{\theta_1, \ldots, \theta_{t-1}\}$ be the observed models at step $t$. Suppose we run Algorithm 1 with $K = 1$ and without rounding, then Algorithm 1 satisfies $(o_\alpha^{(i)} + \frac{\log(1/\delta)}{\alpha-1}, \delta)$-output-specific individual DP for the $i_{th}$ example at $\mathbb{A} = (\theta_1, \ldots, \theta_{t-1}, \mathcal{O}_t)$, where $o_\alpha^{(i)}$ is the accumulated RDP at order $\alpha$ and $\mathcal{O}_t$ is the range of $\mathcal{A}_t$.*

Here we give the proof of Theorem 3.1. Let $(\mathcal{A}_1, \ldots, \mathcal{A}_{t-1})$ be a sequence of randomized algorithms and $(\theta_1, \ldots, \theta_{t-1})$ be some fixed outcomes, we define

$$\hat{\mathcal{A}}^{(t)}(\theta_1, \ldots, \theta_{t-1}, \mathbb{D}) = (\mathcal{A}_1(\mathbb{D}), \mathcal{A}_2(\theta_1, \mathbb{D}), \ldots, \mathcal{A}_t(\theta_1, \ldots, \theta_{t-1}, \mathbb{D})).$$

Noting that the individual RDP parameters of each individual mechanism in $\hat{\mathcal{A}}^{(t)}(\theta_1, \ldots, \theta_{t-1}, \mathbb{D})$ are constants. Further let

$$\mathcal{A}^{(t)}(\mathbb{D}) = (\mathcal{A}_1(\mathbb{D}), \mathcal{A}_2(\mathcal{A}_1(\mathbb{D}), \mathbb{D}), \ldots, \mathcal{A}_t(\mathcal{A}_1(\mathbb{D}), \ldots, \mathbb{D}))$$

be the adaptive composition. In Lemma A.1, we show an RDP bound on $\hat{\mathcal{A}}^{(t)}$ gives an output-specific DP bound on $\mathcal{A}^{(t)}$. We comment that the individual RDP parameters of each individual mechanism in $\mathcal{A}^{(t)}(\mathbb{D})$ are random variables. The composition of random privacy parameters requires additional care because the standard composition theorem requires the privacy parameters to be constants (Feldman & Zrnic, 2021; Lécuyer, 2021; Whitehouse et al., 2022).

**Lemma A.1.** *Let $\mathbb{A} = (\theta_1, \ldots, \theta_{t-1}, \mathcal{O}_t) \subset \mathcal{O}^{(t)}$ where $\theta_1, \ldots, \theta_{t-1}$ are some arbitrary fixed outcomes and $\mathcal{O}^{(t)}$ is the domain of $\mathcal{A}^{(t)}(\mathbb{D})$ and $\hat{\mathcal{A}}^{(t)}(\mathbb{D})$. If $\hat{\mathcal{A}}^{(t)}(\cdot)$ satisfies $o_\alpha$ RDP at order $\alpha$, then $\mathcal{A}^{(t)}(\mathbb{D})$ satisfies $(o_\alpha + \frac{\log(1/\delta)}{\alpha-1}, \delta)$-output-specific differential privacy at $\mathbb{A}$.*

*Proof.* For a given outcome $\theta^{(t)} = (\theta_1, \theta_2, \ldots, \theta_{t-1}, \theta_t) \in \mathbb{A}$, we have $\mathbb{P}\left[\mathcal{A}^{(t)}(\mathbb{D}) = \theta^{(t)}\right] =$

$$\mathbb{P}\left[\mathcal{A}^{(t-1)}(\mathbb{D}) = \theta^{(t-1)}\right] \mathbb{P}\left[\mathcal{A}_t(\mathcal{A}_1(\mathbb{D}), \ldots, \mathcal{A}_{t-1}(\mathbb{D}), \mathbb{D}) = \theta_t | \mathcal{A}^{(t-1)}(\mathbb{D}) = \theta^{(t-1)}\right], \tag{6}$$

$$= \mathbb{P}\left[\mathcal{A}^{(t-1)}(\mathbb{D}) = \theta^{(t-1)}\right] \mathbb{P}\left[\mathcal{A}_t(\theta_1, \ldots, \theta_{t-1}, \mathbb{D}) = \theta_t\right], \tag{7}$$

by the product rule of conditional probability. Apply the product rule recurrently on $\mathbb{P}\left[\mathcal{A}^{(t-1)}(\mathbb{D}) = \theta^{(t-1)}\right]$, we have $\mathbb{P}\left[\mathcal{A}^{(t)}(\mathbb{D}) = \theta^{(t)}\right] =$

$$\mathbb{P}\left[\mathcal{A}^{(t-2)}(\mathbb{D}) = \theta^{(t-2)}\right] \mathbb{P}\left[\mathcal{A}_{t-1}(\theta_1, \ldots, \theta_{t-2}, \mathbb{D}) = \theta_{t-1}\right] \mathbb{P}\left[\mathcal{A}_t(\theta_1, \ldots, \theta_{t-1}, \mathbb{D}) = \theta_t\right], \tag{8}$$

$$= \mathbb{P}\left[\mathcal{A}_1(\mathbb{D}) = \theta_1\right] \mathbb{P}\left[\mathcal{A}_2(\theta_1, \mathbb{D}) = \theta_2\right] \ldots \mathbb{P}\left[\mathcal{A}_t(\theta_1, \ldots, \theta_{t-1}, \mathbb{D}) = \theta_t\right], \tag{9}$$

$$= \mathbb{P}\left[\hat{\mathcal{A}}^{(t)}(\theta_1, \ldots, \theta_{t-1}, \mathbb{D}) = \theta^{(t)}\right]. \tag{10}$$

In words, $\mathcal{A}^{(t)}$ and $\hat{\mathcal{A}}^{(t)}$ are identical in $\mathbb{A}$. Therefore, $\mathcal{A}^{(t)}$ satisfies $(\varepsilon, \delta)$-DP at any $\mathbb{S} \subset \mathbb{A}$ if $\hat{\mathcal{A}}^{(t)}$ satisfies $(\varepsilon, \delta)$-DP. Converting the RDP bound on $\hat{\mathcal{A}}^{(t)}(\mathbb{D})$ into a $(\varepsilon, \delta)$-DP bound with Lemma A.2 then completes the proof.

**Lemma A.2** (Conversion from RDP to $(\varepsilon, \delta)$-DP Mironov (2017)). *If $\mathcal{A}$ satisfies $(\alpha, \rho)$-RDP, then $\mathcal{A}$ satisfies $(\rho + \frac{\log(1/\delta)}{\alpha-1}, \delta)$-DP for all $0 < \delta < 1$.*

$\square$

---

**Algorithm 3** Differentially Private SGD with Individual Clipping

---

**Input:** Clipping threshold $C$, noise variance $\sigma^2$, sampling probability $p$, number of steps $T$.

**for** $t = 0 \; to \; T - 1$ **do**

    **Call Algorithm 2 for individual privacy accounting and get estimates of individual gradient norms $\{Z^{(i)}\}_{i=1}^n$.**

    Sample a minibatch of gradients $\{\boldsymbol{g}^{(I_j)}\}_{j=1}^{|I|}$ with probability $p$ , where $I$ is the sampled indices.

    **Clip gradients $\bar{\boldsymbol{g}}^{(I_j)} = clip(\boldsymbol{g}^{(I_j)}, Z^{(I_j)})$.**

    Update model $\theta_t = \theta_{t-1} - \eta(\sum \bar{\boldsymbol{g}}^{(I_j)} + z)$, where $z \sim \mathcal{N}(0, \sigma^2 \boldsymbol{I})$.

**end for**

---

## B  Individual Privacy Accounting with Individual Clipping

We set $K > 1$ in Algorithm 2 to reduce the computational cost of individual privacy accounting (see Section 3 for details). In this case, the computed privacy costs are estimates of the exact ones. In Section 3.3 we demonstrate the estimates are accurate. In this section, we give another design choice that slightly modifies the original DP-SGD to give exact privacy accounting. More specifically, we clip the individual gradients with the estimates of gradient norms $\{Z^{(i)}\}$ from Algorithm 2. We refer to this design choice as *individual clipping*. We give the implementation in Algorithm 3 and highlight the changes in bold font. We run experiments with individual clipping and report the results in Appendix B.1 and B.2. The experimental setup is the same as that in Section 4.

### B.1  Individual Clipping Does Not Affect Accuracy

Algorithm 3 uses individual clipping thresholds to ensure the computed privacy parameters are strict privacy guarantees. If the clipping thresholds are close to the actual gradient norms, then the clipped results are close to those of using a single maximum clipping threshold. However, if the estimations of gradient norms are not accurate, individual thresholds would clip more signal than using a single maximum threshold.

Table 3: Comparison between the test accuracy of using individual clipping thresholds and that of using a single maximum clipping threshold. The maximum $\varepsilon$ is 7.8 for CIFAR-10 and 2.4 for MNIST.

|  | CIFAR-10 | MNIST |
|---|---|---|
| Individual | 74.0 ($\pm$0.19) | 97.17 ($\pm$0.12) |
| Maximum | 74.1 ($\pm$0.24) | 97.26 ($\pm$0.11) |

We compare the accuracy of two different clipping methods in Table 3. The individual clipping thresholds are updated once per epoch. We repeat the experiment four times with different random seeds. The results suggest that using individual clipping thresholds in Algorithm 1 has a negligible effect on accuracy.

### B.2  Individual Clipping Does Not Change The Observations

Here we show running DP-SGD with individual clipping does not change our observations in Section 4 and 5.

**Privacy parameters have a strong correlation with individual training loss.** In Figure 7, we show privacy parameters computed with individual clipping are still positively correlated with training losses. The Pearson correlation coefficient between privacy parameters and log losses is larger than 0.9 for all datasets.

**Groups are simultaneously underserved in both accuracy and privacy** We show our observation in Section 5, i.e., low-accuracy groups have worse privacy parameters, still holds in Figure 8. We also make a direct comparison with privacy parameters computed without individual clipping. We find that privacy parameters computed with individual clipping are close to those computed without individual clipping. We also find that the order of groups, sorted by the average $\varepsilon$, is exactly the same for both cases.

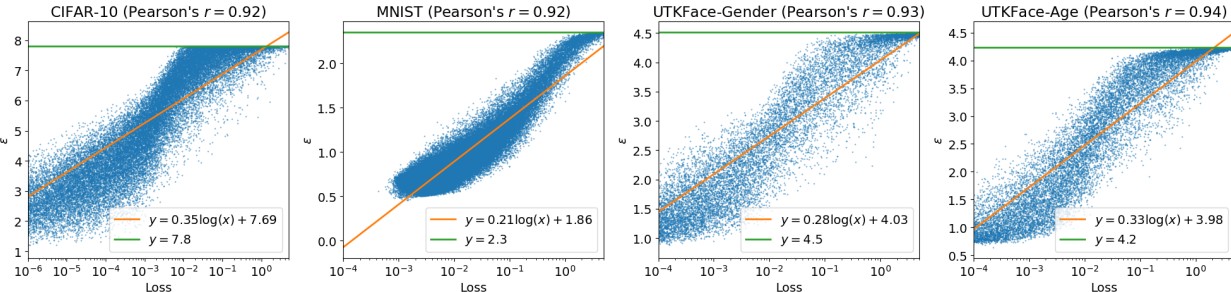

Figure 7: Privacy parameters and final training losses. The experiments are run with individual clipping (Algorithm 3). The Pearson correlation coefficient is computed between privacy parameters and log losses.

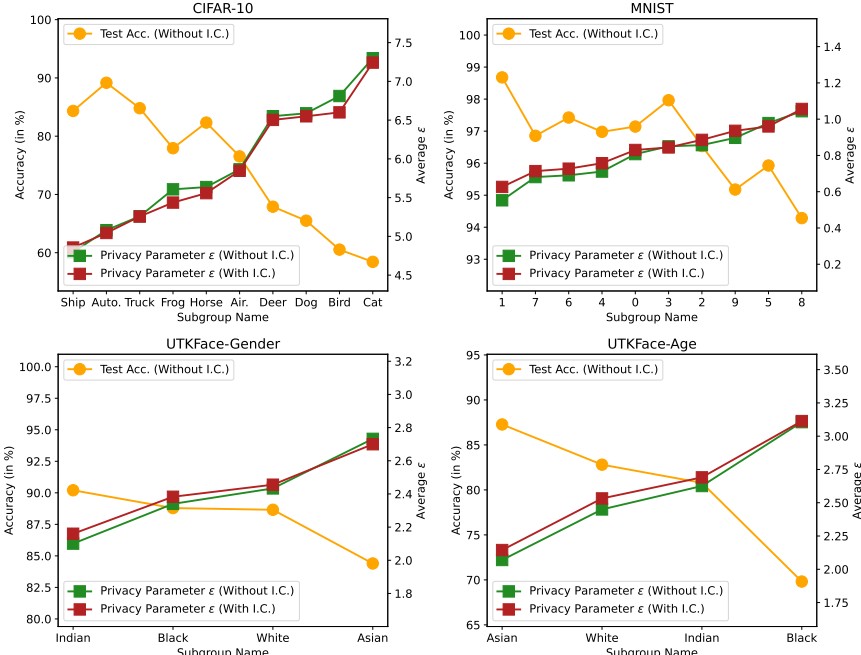

Figure 8: Test accuracy and privacy parameters computed with/without individual clipping (I.C.). Groups with worse test accuracy also have worse privacy in general.

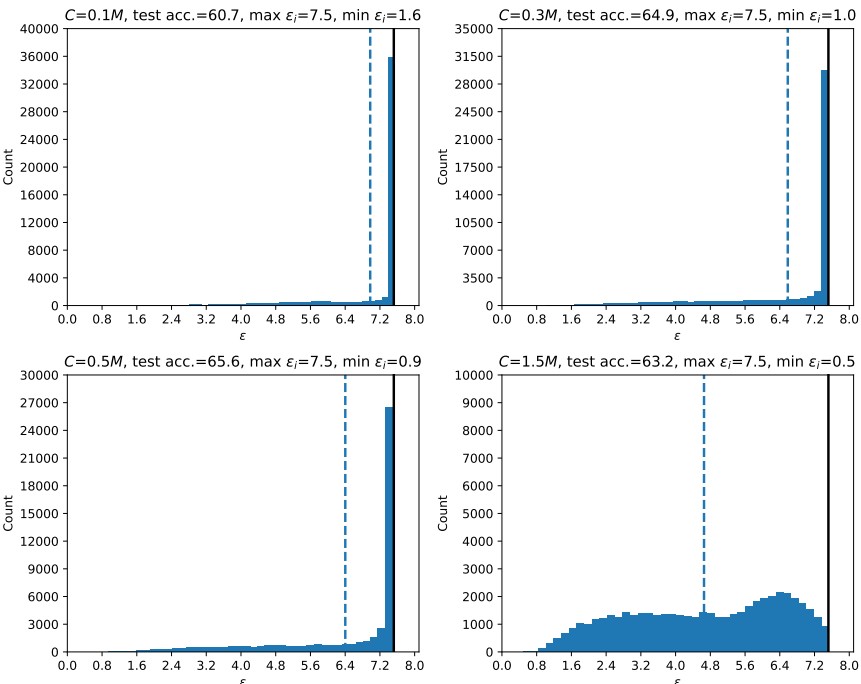

Figure 9: Distributions of individual privacy parameters on CIFAR-10 with different maximum clipping thresholds. The dashed line indicates the average of privacy parameters.

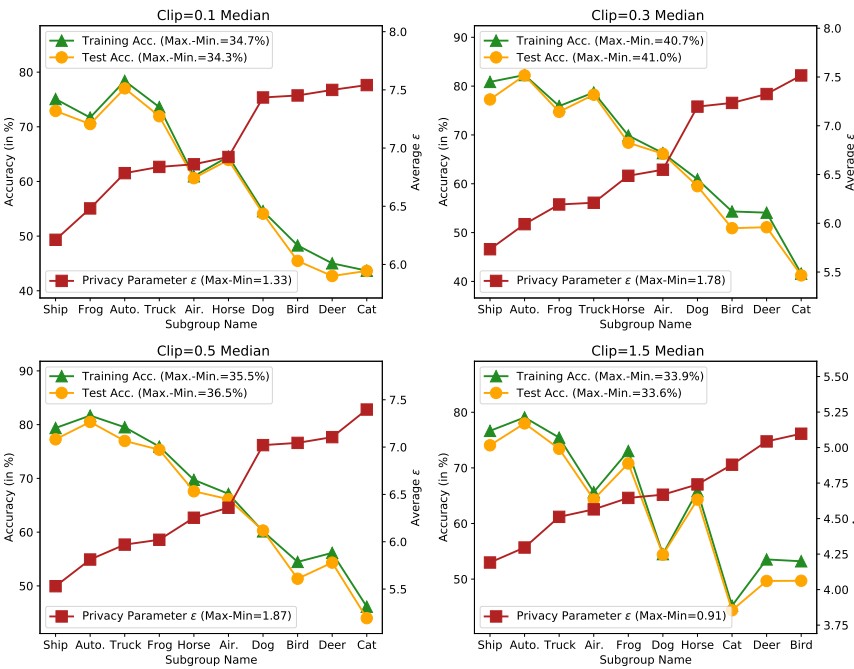

Figure 10: Accuracy and average $\varepsilon$ of different groups on CIFAR-10 with different maximum clipping thresholds.

# C    The Influence of Different Maximum Clipping Thresholds

The value of the maximum clipping threshold $C$ would affect individual privacy parameters. A large value of $C$ would increase the stratification in gradient norms but also increase the noise variance for a fixed privacy budget. A small value of $C$ would suppress the stratification but also increase the gradient bias. Here we run experiments with different values of $C$ on CIFAR-10. We use a small ResNet20 model in He et al. (2016), which only has ∼0.2M parameters, to reduce the computation cost. All batch normalization layers are replaced with group normalization layers. Let $M$ be the median of gradient norms at initialization, we choose $C$ from the list $[0.1M, 0.3M, 0.5M, 1.5M]$.

The histograms of individual privacy parameters are in Figure 9. In terms of accuracy, using clipping thresholds near the median gives better test accuracy. In terms of privacy, using smaller clipping thresholds increases privacy parameters in general. The number of datapoints that reaches the worst privacy decreases with the value of $C$. When $C = 0.1M$, nearly 70% datapoints reach the worst privacy parameter while only ∼2% datapoints reach the worst parameter when $C = 1.5M$.

The correlation between accuracy and privacy is in Figure 10. The disparity in average $\varepsilon$ is clear for all choices of $C$. Another important observation is that when decreasing $C$, the privacy parameters of underserved groups increase quicker than other groups. When changing $C = 1.5M$ to $0.5M$, the average $\varepsilon$ of 'Cat' increases from 4.8 to 7.4, almost reaching the worst-case bound. In comparison, the increment in $\varepsilon$ of the 'Ship' class is only 1.3 (from 4.2 to 5.5).

# D    Privacy Parameters Reflect Empirical Privacy Risks in Non-Private Learning

We run membership inference (MI) attacks to verify whether examples with larger privacy parameters have higher privacy risks in practice. We use a simple loss-threshold attack that predicts an example is a member if its loss value is smaller than a prespecified threshold (Sablayrolles et al., 2019). Previous works show that even large privacy parameters are sufficient to defend against such attacks (Carlini et al., 2019b; Yu et al., 2021). In order to better observe the difference in privacy risks, we also include models trained without differential privacy as target models. For each data subgroup, we use its whole test set and a random subset of the training set so the numbers of training and test loss values are balanced. We further split the data into two subsets evenly to find the optimal threshold on one and report the success rate on another.

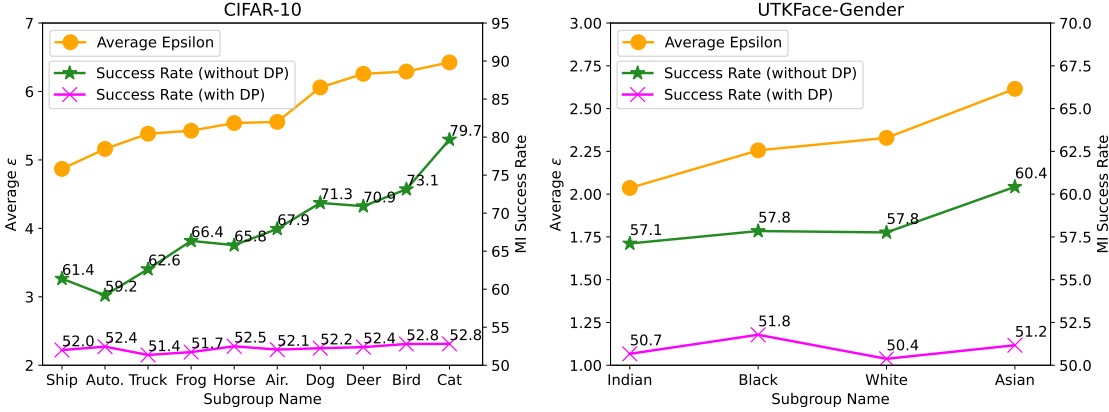

Figure 11: Average $\epsilon$ and membership inference success rates on different subgroups.

The results on CIFAR-10 and UTKFace-Gender are in Figure 11. The subgroups are sorted based on their average $\epsilon$. When the models are trained with DP, all attack success rates are close to random guessing (50%). Although the attack we use can not show the disparity in this case, we note that there are more powerful attacks whose success rates are closer to the lower bound that DP offers (Nasr et al., 2021b). On the other hand, the difference in privacy risks is clear when models are trained without DP. On CIFAR-10, the MI success rate is 79.7% on the Cat class (which has the worst average $\epsilon$ when trained with DP) while is

only 61.4% on the Ship class (which has the best average $\epsilon$). These results suggest that the $\epsilon$ values reflect empirical privacy risks which could vary significantly in different subgroups.

# E The Correlation Between Privacy Parameters and Loss Holds under Different Clipping Thresholds

In Section 4.2, we show that there is a positive logarithmic correlation between privacy parameters and training loss. In this section, we run experiments on CIFAR-10 with $C = 5$ and $C = 10$ to show the correlation still holds under different clipping thresholds. The experiment setup, except the value of $C$, is the same as Section 4.2. The results are in Figure 12. Although changing the clipping threshold changes the slope and intercept, the logarithmic correlation is still strong. The Pearson correlation coefficients are 0.89 and 0.9 for $C = 5$ and $C = 10$, respectively.

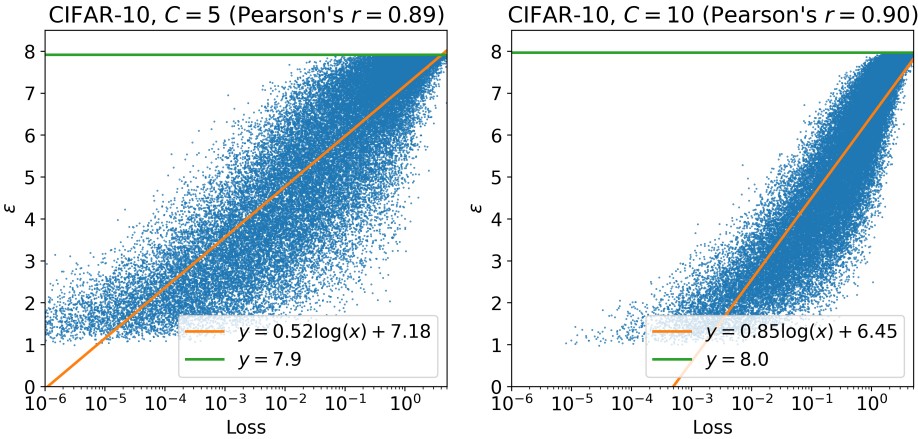

Figure 12: Privacy parameters and final training losses. Each point shows the final training loss and privacy parameter of one example. Pearson's $r$ is computed between privacy parameters and log loss values.

# F Individual Privacy Accounting in More Settings

In this section, we study individual privacy accounting in more experimental settings. The dataset in this section is CIFAR-10. We first study the influence of loss function. We replace the cross-entropy loss with the multi-class hinge loss. Other settings are the same as those in Section 4. The results are in Figure 13. We also study the influence of model architectures. We replace WRN16-4 with the two-layer convolutional neural networks in Papernot et al. (2020) and still use the cross-entropy loss. The learning rate is set as 1.0 and other settings are the same as those in Section 4. The results are in Figure 14.

Our main observations in the main text still hold in the new settings. The Pearson's correlation coefficients between the estimated and actual privacy parameters are larger than 0.99 in both cases. Examples that are underserved by the model also suffer from higher privacy costs. Although the main observations do not change, we observe some differences in the privacy parameters. When using the two-layer neural network instead of WRN16-4, the privacy parameters become larger. For instance, the average $\varepsilon$ of Cat increases from 7.3 to 7.7.

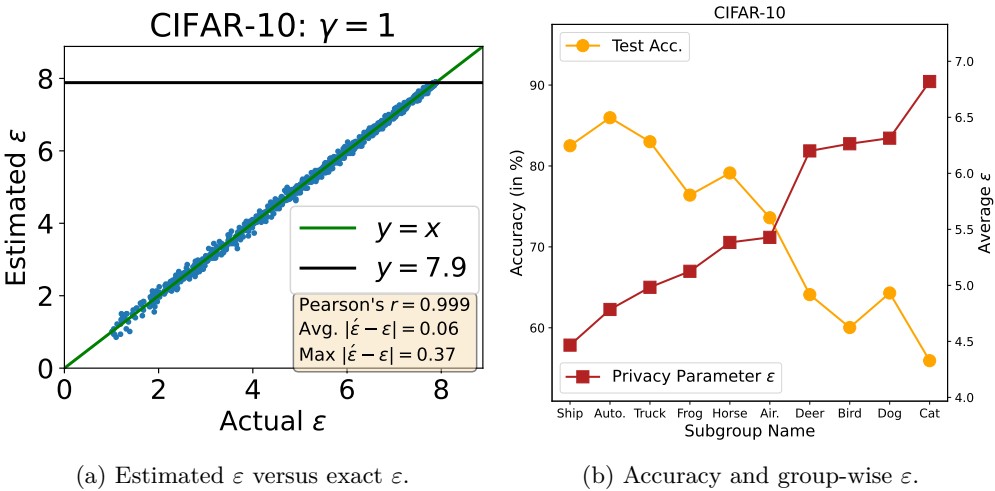

(a) Estimated $\varepsilon$ versus exact $\varepsilon$.

(b) Accuracy and group-wise $\varepsilon$.

Figure 13: The results of using multi-class hinge loss instead of cross-entropy loss.

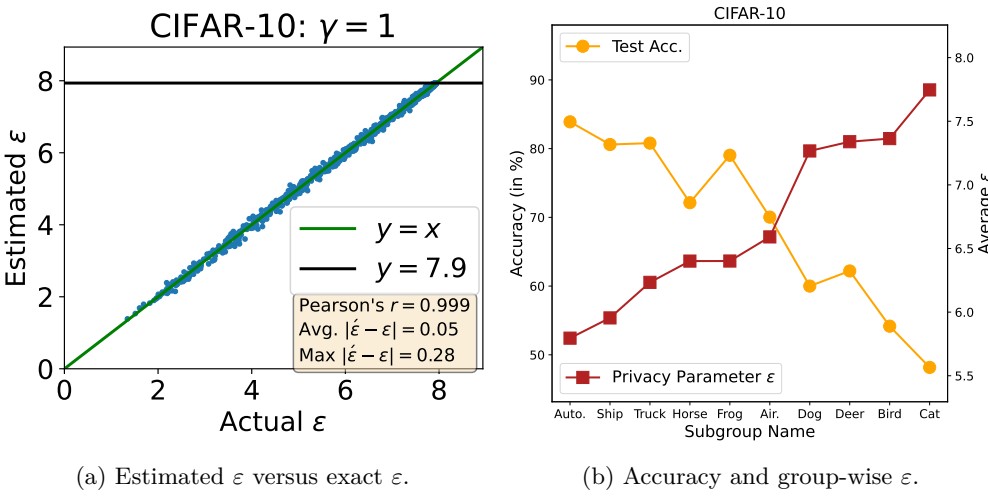

(a) Estimated $\varepsilon$ versus exact $\varepsilon$.

(b) Accuracy and group-wise $\varepsilon$.

Figure 14: The results of using the small convolutional neural network in Papernot et al. (2020).

