# OpenReview forum: "Individual Privacy Accounting for Differentially Private Stochastic Gradient Descent"
_TMLR — Accepted by TMLR_

### Review · Reviewer_EYWv · 2023-05-08

**Summary Of Contributions:**

The paper measures the "individual privacy parameter" obtained for each example from DP-SGD. Intuitively, an example whose gradient norm is not at the clipping threshold will have an individual epsilon value which is smaller than the worst-case epsilon that assumes every example contributes a full-norm gradient. For this setting, the paper proposes an algorithm for computing these individual epsilons, builds a more efficient approximate algorithm, and offers some takeaways on the usefulness of these individual epsilons.

**Audience:**

Yes

**Claims And Evidence:**

Yes

**Requested Changes:**

I would like to see a discussion of the dangers of releasing epsilon to users in the context of a broader system, perhaps where data deletions are an available recourse. (strengthen)

It might be interesting to understand the relationship with individual privacy loss as computed by privacy attacks and this measurement approach to understand how correlated they are. The approach here should be a valid upper bound. (strengthen)

**Strengths And Weaknesses:**

Strengths:
Theorem 3.2 is a very important property for working with these quantities, and I'm glad the authors have addressed this.

The paper offers an analysis of disparate impact in DP that considers both utility and privacy.

The two speedups are nice. It's surprising to me that the privacy accounting step is so slow, but rounding is a nice way to get around that.

Weaknesses:
The direction of privacy filters is quite related to the accounting technique described here, although differences are discussed appropriately in the paper.

Not much fine-grained can be done with these individual epsilons. Even reporting them to the user can result in something nonprivate, if this enables the user to take action in response to learning this. For example, if this prompts a user to issue an unlearning request, this user's deletion can change other users' risks, leaking that the other user had a high epsilon [R1]. Based on the results, user demographics can be correlated with high epsilons, which may lead to disproportionate deletion requests, which can amplify the bias [R2].

Rounding can lead to an underestimation of RDP, if the rounded clipping norm is too small, especially if r is relatively large relative to the true gradient norm.

[R1] - https://arxiv.org/abs/2206.10469
[R2] - https://arxiv.org/abs/1806.08010


Review questions:
Are the claims made in the submission supported by accurate, convincing and clear evidence? I'd say so. All the trends mentioned seem to be quite robust, and the theorems appear correct.

Would at least some individuals in TMLR's audience be interested in knowing the findings of this paper? I think this is of interest to some private ML folks.

---

> ### Author Response · Authors · 2023-05-23
> **Response to Reviewer EYWv**
>
> We thank the reviewer for the comments. We have revised our submission with the following changes. The changes are marked in blue.
>
> **1.Dangers of releasing epsilon to users in the context of a broader system.**
> We add a “Broader Impact” section to discuss the possible implications of combining individual privacy accounting and data deletion.
>
> “One way to utilize individual privacy parameters is releasing them to corresponding users (see Section 3.4 for details). This provides a more accurate, and hence more responsible, privacy report. However, releasing individual privacy parameters may pose new challenges when a machine learning system enables data deletion, also known as machine unlearning (Ginart et al., 2019; Bourtoule et al., 2019). Data deletions may worsen the privacy guarantees of the remaining examples (Carlini et al., 2022). Moreover, groups with larger $\varepsilon$ may send deletion requests more frequently than others, which could  further deteriorate their privacy guarantees (Hashimoto et al., 2018). It's important to keep these considerations in mind when implementing both data deletion and individual privacy accounting in real-world settings. ”
>
> **2.The relationship between individual privacy loss and privacy attacks.**
> We run a simple loss-threshold membership inference attack against trained models and put the results in Appendix D. Below we summarize our findings.
>
> When the models are trained with DP, all attack success rates are close to random guessing (50\%). This aligns with the findings in previous work that even large privacy parameters are sufficient to defend against such attacks (Carlini et al., 2019b). Although the attack we use can not show the disparity in this case, we note that there are more powerful attacks whose success rates are closer to the lower bound that DP offers (Nasr et al., 2021b). When the models are trained without DP, the difference in privacy risks is clear and correlates with the privacy parameters of training with DP. On CIFAR-10, the MI success rate is 79.7\% on the Cat class (which has the worst average $\epsilon$ when the model is trained with DP) while it is only 61.4\% on the Ship class (which has the best average $\epsilon$).
>
>
> **3.Rounding can lead to an underestimation of RDP.**
> In the revised submission, we mention that the rounding $r$ should be small enough to avoid underestimation of RDP. In our experiments, we set $r=0.01C$ which turns out to be enough based on the results in Figure 4.

---

### Review · Reviewer_8AUH · 2023-05-11

**Summary Of Contributions:**

The paper proposes a new DP notion for each individual examples in the dataset. The notion can be used to characterize privacy guarantee for each training sample in DPSGD. For the new notion of DP (which is a generalization of traditional (epsilon, delta)-DP), the authors provided ways to efficiently estimated the privacy cost. One interesting and intuitive observation is that the samples that served poorly by the model also suffer from higher privacy cost.

**Audience:**

Yes

**Broader Impact Concerns:**

No concerns.

**Claims And Evidence:**

Yes

**Requested Changes:**

Have the authors explored how other factors affect the variance of individual privacy cost (e.g., model architecture and loss function as mentioned earlier)? It might be good to try verify the observations in more settings. E.g., how much does model architecture matters?

**Strengths And Weaknesses:**

Strengths:
1. The new individual DP notion is interesting in the sense that it can characterize the privacy guarantee of different training samples in DPSGD
2. The observation that underserved training samples also have high privacy cost is interesting.

Weaknesses:
1. Calculating individual privacy introduces quite significant computation cost in the context of DPSGD.
2. The theoretical contribution is limited since the privacy accounting is straightforwardly using existing techniques.
3. The individual privacy variation could vary depends on datasets, training loss, model architecture, and so on. But the key observation is drawn only from using two datasets. This leaves a question whether the same phenomenon holds if other factors are changed.

---

> ### Author Response · Authors · 2023-05-23
> **Response to Reviewer 8AUH**
>
> We thank the reviewer for the comments. Please find our response below.
>
> **1.Individual privacy accounting in more settings.**
> The model architectures in the initial submission follow the common choices in the literature. We are currently running experiments with different model architectures and loss functions. We will update our submission with the results before the discussion period ends.
>
> **2.Concern about the computation cost of individual privacy accounting.**
> Implementing individual privacy accounting naively introduces significant computational overhead. However, the two speedups in our submission can greatly reduce this overhead and make individual privacy accounting practical.

---

> > ### Author Response · Authors · 2023-05-30
> >
> > Dear Reviewer 8AUH,
> >
> > We study the influence of the loss function by replacing the cross-entropy loss with the multi-class hinge loss, which is a margin-based classification loss function. We also study the influence of model architecture by replacing the WRN16-4 model with the two-layer convolutional neural network in Papernot et al., 2020 [1]. The results have been put in Appendix F of our submission. Our findings are summarized below.
> >
> > Our two main observations still hold in the new settings: 1) The Pearson's correlation coefficients between the estimated and actual privacy parameters are larger than $0.99$ in both cases, suggesting that Algorithm 2 gives accurate estimations of individual privacy, and 2) Examples that are underserved by the model also suffer from higher privacy costs. Although the main observations do not change, we observe some differences in the privacy parameters. For example, when using the two-layer neural network instead of WRN16-4,  the privacy parameters are larger, e.g., the average $\varepsilon$ of the Cat class (CIFAR-10 dataset) increases from 7.3 to 7.7.
> >
> > Regarding the concern about datasets, Figures 4 and 6 in our initial submission also show the results on MNIST and UTKFace-Age. Our observations are consistent across all four tasks we evaluated (MNIST, CIFAR-10, UTKFace-Gender, and UTKFace-Age).
> >
> > [1]: Nicolas Papernot, Abhradeep Thakurta, Shuang Song, Steve Chien, and Úlfar Erlingsson. Tempered Sigmoid Activations for Deep Learning with Differential Privacy. https://arxiv.org/abs/2007.14191.
> >
> > Authors of Paper 1085

---

### Review · Reviewer_9Tbq · 2023-05-20

**Summary Of Contributions:**

This paper studies DP-SGD and introduces a “per-example privacy guarantee”, a definition achieved by specific runs of the algorithm on each sample for a specific datasets. The paper first claims that a large proportion of examples benefit from a stronger privacy guarantee. Second, that per-class accuracy is correlated with the level of privacy, i.e., small privacy budget $\epsilon$ on a given class correlates with higher test accuracy for that class.

**Audience:**

Yes

**Broader Impact Concerns:**

The paper makes claims on the privacy risk of users. I don’t think there are ethical concerns, but rather incorrect claims and/or misleading interpretations of results that could lead to leaking private information.


**Claims And Evidence:**

No

**Requested Changes:**

1. Most DP-SGD implementations use shuffle data, not poisson sampling. However, I understand that most of the literature uses this simplification because privacy accounting is simpler under this assumption. I would encourage the authors to include a note on this for transparency.


2. The privacy definition of being “individual” is misleading since it depends on the dataset. Assume the algorithm is learning a binary classification model on a dataset $D_1=[(x_1, y_1), …, (x_n, y_n)]$ where  $ x_1\approx x_2\approx...\approx x_n $ , and $y_i=0$. After a few training rounds it is likely that the gradient at $x_1$ will be small since it already “learned” the information from $x_1$ through other samples, and will have a small budget.
However, if we had a dataset $D_2=[(x_1, y_1), (w_2,z_2), …, (w_n, z_n)]$,  where $w_i = -x_i$, and $z_2=...=z_n=1$, in this case the gradient at $(x_1,y_1)$ might be large since it is learning new information from what it learns from other data points, and consequently its budget might be much larger.


3. Theorem 3.2 is incorrect when assuming $f$ is only post-processing since $Z_i$ depends on the non-private norm at iteration $t$ and this norm itself depends on the whole dataset.

4. Revealing privacy budgets reveals information to user $i$ about other users. This step has to be clarified.

5. DP-SGD computes per-example gradient norms for clipping at each iteration (line "`clip gradient norms g^(I_i)...`"  in algorithm 1). Efficiency of this step is a large area of research and recently `ghost` clipping techniques try to devise efficient ways of computing these norms. Why would norms have to be recomputed in algorithm 2? Maybe something needs clarification or sections 3.1. and 3.2. are redundant since norms can be reused from this step.

6. Wouldn’t clipping to Z(i), as suggested in section 3.2 mean no clipping at all?

7. In the experiments the authors define “C as the median of gradient norms at initialization,”. This step does not respect privacy unless the median is privatized.

8. “Intuitively, an example would have a high loss after training if its gradient norms across training are large” I don’t see how this is intuitive. Large steps could result precisely in an optimal point with minimum loss.  Additionally, the relation of the loss value with the clipping parameter is not discussed. Large privacy budgets being correlated with high loss values could be due to over-clipping.

**Strengths And Weaknesses:**

*Strengths*

Privately learning deep learning models is an active field of research and DP-SGD is widely used. This paper aims at advancing the understanding of privacy guarantees and its implications for specific users, in terms of exposure risk and accuracy which I think is of interest to the community.

*Weaknesses*

- The paper makes several mistakes on privacy accounting that can lead to private information being leaked (see requested changes).
- The suggested “individual” privacy definition is misleading since individual budgets depend on all other data points on the dataset. Specifically, the budget of sample $x$ on the dataset $[y_1, …, y_n, x]$ can be drastically different from its budget if the model is learned from the dataset $[z_1, …, z_n, x]$.
- I think there is an inconsistency between algorithm 1 and the implementation used by the authors regarding computation of per-example gradients norms. Algorithm 1 computes the gradients for the clipping step so it is unclear to me the section spent on avoiding computation of gradients norms for privacy accounting in section 3.1.

---

> ### Author Response · Authors · 2023-05-23
> **Response to Reviewer 9Tbq (Part 1/2)**
>
> We thank the reviewer for the detailed comments. In the response below, we first explain why Theorem 3.2 is correct. Then we respond point-to-point to the reviewer’s other concerns. We have revised our submission accordingly and the changes are marked in blue.
>
> **1.Correctness of Theorem 3.2.**
> The full input to the function $f_{i}(\cdot)=f(\cdot,d_{i})$ is $(\theta_{1},\ldots,\theta_{t-1},d_{i})$. The value of $Z(i)$ at step $t$ is a function of $(\theta_{t-1},d_{i})$. Although $Z(i)$ depends on the whole dataset, the dependence is from $\theta_{t-1}$ which is released in a privacy-preserving manner. Therefore, for $d_{j}\neq d_{i}$, $f_{i}$ is a post-processing function of $(\theta_{1},\ldots,\theta_{t-1})$. The correctness of Theorem 3.2 also explains why revealing $\varepsilon_{i}$ only to user $i$ does not increase the privacy cost of other users.
>
> **2.Most DP-SGD implementations use shuffle data.**
> We add a footnote at the beginning of Section 3.
>
> “Our implementation of DP-SGD follows the privacy analysis in Abadi et al., 2016 that uses Poisson sampling. We note that many existing implementations of DP-SGD use shuffle data instead of Poisson sampling to enforce stochasticity. Shuffle data is easier to implement but using it would create a mild discrepancy with the analysis in Abadi et al., 2016. Formal privacy analysis of shuffle data requires \emph{privacy amplification by shuffling} (Koskela et al., 2023; Wang, 2023; Feldman et al., 2023). ”
>
> **3.The privacy definition depends on the dataset.**
> Privacy-preserving ML is about sharing the patterns in the dataset while withholding the information of any individual. Therefore, it is true that an individual consumes more privacy budget if it is an “outlier” with respect to the training set. The proposed output-specific DP notion aims to describe this intuition in a more rigorous way. We argue that it is reasonable for an individual privacy definition to depend on the dataset. If a definition is dataset-agnostic, one has to assume the worst-case dataset and hence the worst-case privacy bound.
>
> In the revised version, we add the following line in the introduction to further clarify that individual privacy in this paper depends on the training set.
>
> “The trajectory of one DP-SGD run, and hence the individual privacy in this paper, is impacted by various factors such as the training set and the privacy noise.”
>
> **4.Why would norms have to be recomputed in Algorithm 2?**
> In Algorithm 1, only the gradient norms of the current minibatch are computed. To compute the privacy cost of an example at step $t$, we need its gradient norm even if it is not sampled in the current minibatch. This is because of the analysis of privacy amplification by subsampling. In the formulation of the analysis, which we restate in Eq (1) and (2), every example has a probability $p$ of being sampled. Therefore, we need to compute full batch gradient norms in Algorithm 2. Moreover, we can not reuse the mini-batch gradient norms in Algorithm 1 because they depend on the stochasticity of Poisson sampling. Keeping the secrecy of such stochasticity is essential for the analysis of privacy amplification by subsampling.
>
> **5.Wouldn’t clipping to Z(i)  mean no clipping at all?**
> In Appendix B.2, we use Z(i) to clip per-example gradients. Whether the clipping happens is based on the choice of $K$, which is one input argument of Algorithm 2. When $K=1$, there is indeed no clipping at all. When $K>1$, there might be clipping because the gradient norm of one example may vary at different iterations. In our experiments, we set $K>1$ to speed up the computation. Figure 4 shows that using $K>1$ still yields accurate estimations of individual privacy parameters.
>
> **6.Setting C as the median of gradient norms at initialization does not respect privacy.**
> On CIFAR-10, we set C as a constant. On UTKFace and MNIST, we set C as the median of gradient norms at initialization.  The medians of UTKFace and MNIST are indeed not privatized. We have clarified this in our submission. This design choice follows Abadi et al., 2016 (see their discussion on the clipping bound in Section 5.2 of Abadi et al., 2016).
>
> To show that the median of gradient norms could be released accurately with a small privacy cost. We use the algorithm in Andrew et al., 2021 to privatize the median of the UTKFace-Gender dataset. We set $(\varepsilon=0.1,\delta=1\times 10^{-5})$. The non-private median is 15.73 and the privatized median is 15.82.

---

> ### Author Response · Authors · 2023-05-23
> **Response to Reviewer 9Tbq (Part 2/2)**
>
> **7.“An example would have a high loss after training if its gradient norms across training are large.” is not intuitive.**
> Our intuition is based on the fact that in strongly convex optimization, the loss value of an example is reflected in the norm of its gradient. Thus, a larger gradient norm could indicate a larger loss. We have revised our discussion as follows.
>
> “In strongly convex optimization, the loss value of an example is reflected in the norm of its gradient. However, for non-convex deep models, there is no clear relation between the final training loss of one example and its gradient norms. Therefore, we run experiments to reveal the empirical correlation between privacy and utility.”
>
>
> **8.Large privacy budgets being correlated with high loss values could be due to over-clipping.**
> In our initial submission, we set $C=1$ on CIFAR-10 following the setup in De et al., 2022. We are running experiments with $C=5$ and $C=10$ on CIFAR-10 and will update the results before the discussion period ends.

---

> > ### Author Response · Authors · 2023-05-30
> >
> > Dear Reviewer 9Tbq,
> >
> > We have updated our submission with experiments using larger clipping thresholds (Figure 12 in Appendix E).  The Pearson correlation coefficients are $0.89$ and $0.9$ for $C=5$ and $C=10$, respectively, suggesting that there is still a positive logarithmic correlation.
> >
> > Authors of Paper 1085

---

### Decision · Action_Editors · 2023-08-03

**Recommendation:** Accept as is

**Comment:**

The paper proposes an example specific notion of epsilon, delta DP, where the epsilon is defined for a an example, outcome pair. The paper claims that a large proportion of examples in a training during DP-SGD benefits from this more nuanced notion. The reviewers found this notion interesting and useful. There were some clarifications/corrections requested by the reviewers which were answered by the authors. Reviewers had the general concern that the individual privacy guarantees usefulness is limited due to the risk of exposure however they unanimously leaned towards acceptance.

**Audience:**

The primary audience for the paper will be researchs working in differential privacy. The reviewers found the paper to be of sufficient novelty in this regard.

**Claims And Evidence:**

The reviewers found the claims of the submission supported by accurate and clear evidence.